# Long Circulating RNAs Packaged in Extracellular Vesicles: Prospects for Improved Risk Assessment in Childhood B-Cell Acute Lymphoblastic Leukemia

**DOI:** 10.3390/ijms26093956

**Published:** 2025-04-22

**Authors:** Lucas Poncelet, Chantal Richer, Angela Gutierrez-Camino, Teodor Veres, Daniel Sinnett

**Affiliations:** 1Division of Hematology-Oncology, CHU Sainte-Justine Research Center, Montreal, QC H3T 1C5, Canada; lucas.poncelet@cnrc-nrc.gc.ca (L.P.); chantal.richer.hsj@ssss.gouv.qc.ca (C.R.); angela.gutierrez@ehu.eus (A.G.-C.); 2Medical Devices Research Centre, National Research Council Canada, Boucherville, QC J4B 6Y4, Canada; teodor.veres@cnrc-nrc.gc.ca; 3Pediatric Oncology Group, BioBizkaia Health Research Institute, 48903 Barakaldo, Spain; 4Department of Pediatrics, Faculty of Medicine, University of Montreal, Montreal, QC H3T 1J4, Canada

**Keywords:** extracellular vesicles, circulating RNAs, biomarkers, childhood leukemia

## Abstract

Analysis of tumoral RNA from bone marrow (BM) biopsy is essential for diagnosing childhood B-cell acute lymphoblastic leukemia (B-ALL), risk stratification, and monitoring, by detecting fusions and gene expression patterns. However, frequent BM biopsies are invasive and traumatic for patients. Small extracellular vesicles (sEVs) circulating in blood contain a variety of biomolecules, including RNA, that may contribute to cancer progression, offering a promising source of non-invasive biomarkers from liquid biopsies. While most EV studies have focused on small RNAs like microRNAs (miRNAs), the role of longer RNA species, including messenger RNAs (mRNAs), long non-coding RNAs (lncRNAs), and circular RNAs (circRNAs), remains underexplored despite their demonstrated potential for risk-based patient stratification when starting from BM biopsies. We used immuno-purification to isolate sEVs from peripheral blood at diagnosis in B-ALL patients and cell model-based conditioned culture medium (CCM) with *ETV6::RUNX1* and *TCF3::PBX1* fusions. Using whole-transcriptome sequencing targeting transcripts over 200 nt and a novel data analysis pipeline, we identified 102 RNA transcripts (67 mRNAs, 16 lncRNAs, 10 circRNAs, 4 pseudogenes, and 5 others) in patient-derived sEVs. These transcripts could serve as biomarkers for two distinct molecular subgroups of B-ALL, each with different risk profiles at diagnosis. This is the first study characterizing the long transcriptome in blood-derived sEVs for childhood B-ALL, highlighting the potential use of circulating RNAs for improved risk-based stratification.

## 1. Introduction

Childhood B-ALL is the most common type of pediatric cancer [1], accounting for about 25% of all cancer cases in children. B-ALL is a heterogeneous disease with subtypes defined by chromosomal/molecular alterations, each with different prognoses at the time of diagnosis. Patients carrying *ETV6::RUNX1* translocation account for approximately 20% of cases and have a good prognosis although with some potential for relapse [2,3] while patients presenting with the *TCF3::PBX1* fusion gene, which account for approximately 3–5% of cases, have a poorer prognosis [4,5]. In the current standard of care, subtype-based risk stratification and monitoring of B-ALL is already well implemented in clinical practice through cytogenetics, and molecular biology [2] from BM biopsies. In recent years, total RNA sequencing (RNA-seq) is emerging as a suitable alternative to those well-implemented methods for risk stratification, as it allows B-ALL subtyping by detecting expression profiles of coding genes, fusion genes, and sequence mutations in a single assay in a clinically relevant turnaround time [6,7]. LncRNA) expression [8,9,10], and circRNA expression [11,12] in tumors have also been assessed for risk stratification of B-ALL patients. However, the invasive and traumatic nature of BM biopsy procedures pose significant challenge and discomfort for patients, limiting their frequency [13]. Subsequent analysis is also time consuming, further highlighting the need for novel and accessible biomarkers for patient risk stratification. One proposed solution is to use peripheral blood (PB)-derived biomarkers as a liquid biopsy [14,15].

RNA transcripts can be exported to biofluids [16,17], where miRNAs are bound to protein complexes or encapsulated in extracellular vesicles (EVs) [18,19], while long RNAs are mostly encapsulated in EVs to protect them from RNAse degradation [20,21]. Different methods are used for EVs isolation, with ultracentrifugation (UC) being the gold standard [22]. These isolation techniques are often limited by large sample requirements, which is particularly challenging in the context of pediatric patients with already low blood volumes available for clinical and research purposes. Size exclusion chromatography (SEC) is a better alternative for 0.5–1 mL volume ranges [23,24] and further purification with magnetic beads has been shown to maximize the potential for biomarker discovery [25]. While most studies looking at EVs content and/or circulating RNAs in peripheral blood in the context of childhood leukemia are focused on miRNAs [26,27,28,29,30], long RNA transcripts, defined here as RNA molecules longer than 200 nucleotides (nt), including mRNAs, lncRNAs, and circRNAs, can also be found in sEVs, also called exosomes [31,32,33]. However, little is known about long RNA transcripts shuttled by sEVs suspended in the peripheral blood of childhood B-ALL patients and their potential use as circulating biomarkers.

Therefore, we aimed to comprehensively characterize the long RNA content, including mRNAs, lncRNAs, and circRNAs, within sEVs isolated from the plasma of childhood B-ALL patients at diagnosis, as well as from CCM of their representative B-ALL cell models. We focused on two distinct molecular subtypes with different risk profiles: *ETV6::RUNX1^+^* (lower risk) and *TCF3::PBX1^+^* (higher risk) childhood B-ALL. We developed a novel model-based data-analysis pipeline to uniformly quantify most linear and circular transcript biotypes from total RNA-seq data. Using this approach, we observed differences in RNA biotype distribution between blood and cell-derived sEVs, compared with corresponding BM biopsies and whole-cell extracts. Looking at patient-blood-derived IM-sEVs, we identified 102 potential circulating biomarkers for B-ALL subtypes (70 *ETV6::RUNX1*-specific, 17 *TCF3::PBX1*-specific, and 15 shared between them), encompassing mRNAs, lncRNAs, circRNAs, and pseudogenes. In a parallel analysis of BM biopsies alone, we did not observe subtype-specific variations for these 102 circulating RNAs, suggesting the presence of a useful signature in EVs for subtype-based risk assessment, although different from the one present in the bone marrow. To explore potential functionalities linked to these EV-shuttled transcripts, we performed correlation analyses on BM-derived datasets. Our comprehensive examination of the long transcriptome in blood cells and sEVs provides new perspectives for non-invasive, risk-based assessment and stratification of B-ALL patients at the time of diagnosis.

## 2. Results

### 2.1. Comparative Analysis of sEV Subfractions: Size, Protein Composition, and RNA Content

Small EVs (sEVs) were purified from CCM of B-ALL cell lines and B-ALL patient plasma (Figure 1A,B). sEVs retained after SEC purification and 20,000× *g* centrifugation exhibited a particle size distribution ranging from 60 nm to 130 nm (Figure 1C). Subsequent purification of these sEVs using immuno-affinity resulted in an enriched population of IM-sEVs, which were characterized by higher levels of CD63, CD81, CD9, and Synt-1 proteins as compared to sEVs purified using SEC only, together with minimal presence of Cyt-C and lipoproteins Apo-A1 and Apo-E (Figure 1D and Appendix A). The IM-sEV subfraction was almost completely depleted from lipoproteins as compared to unprocessed plasma input samples (Appendix A). Starting with aliquots from the same input sample, the IM-sEV subfraction yielded a lower amount of RNA after extraction, compared to all sEVs which contained, on average, 1.2 times more RNA than IM-sEVs (Figure 1E). Both vesicle types showed small RNA fragment enrichment, and also contained a wide range of transcript sizes, as shown by the DV200 values, representing the percentage of RNA fragments longer than 200 nt in the sample. When studying EV-derived samples, DV200 and RNA integrity number (RIN) values are used in tandem to assess overall RNA quality. For cell-line-derived EVs, the overall RNA content in the IM-sEVs fraction (RIN~6.6 and DV200~72%) exhibited higher quality and less degradation compared to the sEVs fraction (RIN~2.3 and DV200~40%). For IM-sEVs RNA derived from patient plasma, we obtained an average of RIN~2.2 and DV200~30% (Figure 1F). Following library preparation, whole-transcriptome sequencing, and pseudo-alignment to a transcriptome reference, the IM-sEVs fraction exhibited an average read mapping rate of 24% compared to 1.5% for sEVs (Figure 1G). These findings suggest that size- and affinity-purified IM-sEVs contain RNA of similar or higher integrity compared to sEVs isolated by SEC only and may be better suited for RNA-seq.

### 2.2. Integrating circRNAs, lncRNAs, and Fusion Gene Transcripts for Extensive RNA Biotypes Quantification

In addition to all the genes annotated in Gencode, we sought to integrate circRNAs, additional reported lncRNAs, and fusion gene transcripts into a single reference for quantification of all biotypes within a single software (Figure 2A). By combining the results obtained from two functionally distinct detection tools, CIRI2 and Circexplorer2, and removing duplicate entries, we detected a total of 102,302 exonic circRNA transcripts corresponding to 11,367 genes as well as 1843 non-exonic circRNA transcripts (Figure 2B). In addition to the 15,006 lncRNA transcripts in the Gencode V29 annotation, we incorporated 39,074 lncRNA transcripts exclusive to the LNCipedia 5.2 database (Figure 2C). Using Arriba and STAR-Fusion, we included four transcripts for the *TCF3::PBX1* fusion gene, three transcripts for the *ETV6::RUNX1* fusion gene, and three transcripts for the reciprocal *RUNX1::ETV6* fusion gene. We verified that the expression of the fusion transcripts was only observed in samples known to carry the corresponding genetic alteration (Figure 2D and Appendix A). We constructed an extensive reference to accurately detect and quantify a wide range of RNA biotypes potentially encapsulated in blood sEVs and suspected to play subtype-specific roles in B-ALL.

### 2.3. Characterization of the Long Transcriptome in Circulating IM-sEVs of Childhood B-ALL Patients

We assembled a cohort of 24 childhood B-ALL patients equally distributed into cases carrying the *ETV6::RUNX1* or the *TCF3::PBX1* fusion genes (Table 1). As healthy B-lymphocyte controls, we included three B-cell samples consisting of CD10+/CD19+ blood cord cell samples. This cohort allowed us to investigate RNA export in IM-sEVs circulating in the peripheral blood of childhood B-ALL patients and gene expression patterns in related BM biopsies. A subset of 10 patients (5 *ETV6::RUNX1^+^* and 5 *TCF3::PBX1^+^*) had both total tumoral RNA and total peripheral blood IM-sEVs RNA sequenced. Cellular models of *ETV6::RUNX1^+^* (REH) and *TCF3::PBX1^+^* (697) were also used for protocol optimization and transcriptome analysis.

Using a threshold of two transcripts per million (TPM), we identified the expression of 16,581 genes in REH cells, 9636 in REH-derived IM-sEVs, as well as 15,450 in 697 cells, and 8436 in 697-derived IM-sEVs (Figure 3A). Notably, 600 genes were exclusively detected in REH IM-sEVs and 421 in 697 IM-sEVs (Appendix A). A total of 116 genes were shared between IM-sEVs derived from both cell lines (Appendix A). In patients’ samples, we observed the expression of 10,528 genes in *ETV6::RUNX1^+^* tumors, 1315 in *ETV6::RUNX1^+^* IM-sEVs, 14,247 in *TCF3::PBX1^+^* tumors, and 801 in *TCF3::PBX1^+^* IM-sEVs (Figure 3B). Among these, 617 genes were only detected in *ETV6::RUNX1^+^* IM-sEVs, and 414 in *TCF3::PBX1^+^* IM-sEVs (Appendix A). We observed a large proportion of overlapping genes (between 70% and 94%) for whole-cell extracts and tumor samples, as well as CCM-derived IM-sEVs (Figure 3A,B). However, looking at blood-derived IM-sEVs, the proportion of overlapping genes is less than 10% (Figure 3B). For both subtypes, we also observe that the number of overlapping genes for all fractions (5823 for cells and CCM-derived IM-sEVs vs. 70 for tumors and blood-derived IM-sEVs) is greater in cell lines than in patient samples (Appendix A). Overall library composition was significantly different between cellular models, tumor samples, controls (Figure 3C), and corresponding IM-sEVs fractions (Figure 3D). The libraries derived from IM-sEVs exhibited a higher representation of lncRNAs and circRNAs biotypes. Furthermore, the analysis of genes differentially expressed in IM-sEVs compared to tumors (log2 fold change > 1, adjusted *p*-value < 0.05) emphasized the enrichment of lncRNAs, circRNAs, pseudogenes, and a moderate number of mRNAs in IM-sEVs (Figure 3E). The fusion transcripts *ETV6::RUNX1* and *TCF3::PBX1* were detected in patient-BM-derived tumoral RNA-seq data but not in blood-derived IM-sEVs (Appendix A). The absence of both fusion transcripts in blood-derived IM-sEVs was also confirmed by RT-qPCR (Appendix A). RNA composition in blood IM-sEVs differs from tumoral samples and B-cell controls with an enrichment in ncRNA biotypes. A complete list of genes detected in patients and cell lines for a TPM > 2 is available in Appendix A.

### 2.4. Identification and Validation of Circulating RNA Biomarker Candidates in Childhood B-ALL IM-sEVs

Considering only genes that have RNA-seq reads present in at least 2 samples within each subtype (2/5), we obtained a list of 70 genes for *ETV6::RUNX1^+^* IM-sEVs (n = 5) and 17 genes for *TCF3::PBX1^+^* IM-sEVs (n = 5). Of those, there were 15 overlapping genes between the two subtypes (Figure 4A, Appendix A). We conducted quantitative RT-qPCR analysis on 16 RNA biomarker candidates out of the 102 identified in IM-sEVs from ETV6::RUNX1^+^ and *TCF3::PBX1^+^* patients (Figure 4B). For highly expressed genes, we observed similar differential expression patterns between B-ALL subtypes in RNA-seq and RT-qPCR data. Genes detected with lower read counts in our RNA-seq dataset like lncRNAs DANT1 and lnc-EGFR-4 and mRNA TCF19 were not detectable by RT-qPCR. We then conducted differential expression analysis on BM-derived RNA-seq datasets only (12 *ETV6::RUNX1^+^* tumors versus the 12 *TCF3::PBX1^+^* tumors). We observed no significant differences in expression levels in tumor samples for the 102 putative RNA circulating biomarker candidates, except for the PRKCB gene, coding for protein kinase C Beta (Appendix A). We also conducted differential expression analysis on BM-derived tumors versus B-cell control RNA-seq datasets. We found that 9 candidates exhibited significantly higher expression in tumors compared to blood-cord-derived B-cell controls, while 14 candidates displayed significantly higher expression in controls (Appendix A). Although the 102 B-ALL RNA biomarker candidates did not exhibit consistent differential expression in tumoral transcriptomes, we observed their selective export into the IM-sEVs compartment in a subtype-specific manner.

### 2.5. Circulating IM-sEVs Biomarker Candidates Are Linked to Childhood B-ALL and Its Subtypes

We conducted a weighted gene co-expression network analysis (WGCNA) using two distinct datasets. We compared 24 B-ALL tumors (combining both *ETV6::RUNX1^+^* and *TCF3::PBX1^+^* patients BM biopsies) with 3 CD10+/CD19+ B-cell controls. We also compared the 12 *ETV6::RUNX1^+^* tumors versus the 12 *TCF3::PBX1^+^* tumors. For each dataset, the top 5000 variable genes encompassing the 102 potential RNA biomarker candidates were organized into modules according to their expression patterns. Within these modules, we specifically examined the presence and behavior of the 102 candidates to delve deeper into their potential associations with B-ALL, as well as any subtype-specific connections. Analysis of the first dataset yielded 12 modules (A1 to A12) correlated with disease status (B-ALL vs. healthy controls). Among the 102 potential RNA biomarker candidates, 21 transcripts were located in modules exhibiting positive correlation with the B-ALL status (A5, A7, A10, A11). Notably, 10 out of the 21 were located in the A7 and A10 modules, demonstrating the strongest association with B-ALL (Figure 5A). In the second dataset, we identified nine modules (B1 to B9). A total of 51 of the 102 biomarker candidates were located in modules correlated with disease subtype (*ETV6::RUNX1^+^* vs. *TCF3::PBX1^+^*). Notably, the lncRNA lnc-GSDME3 and the mRNA PRKCB were located in the B3 module, which showed a strong correlation with the *ETV6::RUNX1^+^* subtype. The mRNAs C1orf115 and GPM6A were located in the B6 module, showing a strong correlation with the *TCF3::PBX1^+^* subtype (Figure 5B). All candidate genes located in modules that significantly correlated with either disease or subtype are listed in Appendix A. To gain insight into the biological functions and pathways associated with disease status or B-ALL subtype, we performed gene ontology (GO) and Kyoto Encyclopedia of Genes and Genomes (KEGG) pathway enrichment analyses for the modules of interest (A7, A10, B3, and B6). We observed significant associations with cellular pathways and mechanisms related to cancer (Figure 5C,D and Appendix A). Modules A7 and A10, correlated with B-ALL, are enriched in genes related to mitosis, DNA replication and repair, cytokinesis, and G2/M transition, all hinting at mechanisms of active cell division. Modules B3 and B6, related to subtype-specific traits, are associated with cellular mechanisms, but with no clear link to cancer (bone morphology, nervous system). We also performed a functional gene set enrichment analysis (FGSEA) using the MSigDB hallmark gene set collection (Figure 5E). Out of 50 Hallmark gene sets in the complete collection, 33 were significantly correlated with the expression of the 102 potential RNA biomarker candidates. Overall, the *ETV6::RUNX1*-related candidates exhibited a significant association (FDR < 0.05) with a higher number of gene sets in the collection (21 ± 6 sets) compared to the *TCF3::PBX1*-related candidates (11 ± 4 sets). Within the *ETV6::RUNX1*-related candidates, we identified two main clusters (A and B, Figure 5E), primarily distinguished by their correlation with three gene sets related to MYC targets, oxidative phosphorylation, and E2F targets (Appendix A). For the *TCF3::PBX1*-related candidates, we also identified two main clusters (C and D, Figure 5E), related to heme metabolism, MTORC1 signalling, MYC targets, protein secretion, G2M checkpoint, and epithelial mesenchymal transition (Appendix A). These findings point to the functional roles and pathways associated with the identified RNA biomarker candidates and shed light on the potential mechanisms underlying the disease and its subtypes.

## 3. Discussion

In this study, we collected both bone marrow biopsies and blood-derived IM-sEVs from B-ALL patients and used high-throughput genomics to build a comprehensive understanding of the gene expression landscape and RNA export in EVs for two tumor subtypes. We evaluated the potential of analysing long RNA transcripts encapsulated in blood-derived IM-sEVs as non-invasive biomarkers. We identified 102 circulating RNA candidates (67 mRNAs, 16 lncRNAs, 10 circRNAs, 4 pseudogenes, and 5 others) within the 2 molecular subgroups of childhood B-ALL patients. Our findings highlight that long RNA content shuttled by IM-sEVs suspended in B-ALL patient blood can provide novel insights and lead to the discovery of RNA-based biomarkers positively linked with subtype-specific characteristics.

Building on previous work [31], we investigated the long RNA landscape in the tumoral and vesicular compartments using a novel bioinformatic approach to look at mRNAs, lncRNAs, circRNAs, and fusion transcripts. Due to very limiting constraints on material availability, we used short-read-based low-input RNA-sequencing. As RNA is fragmented during the library preparation process, all transcripts are linearized. Linear RNA transcripts are the easiest to quantify via simple read counting that map to their sequences. As both linear and circular RNAs share the same exonic sequences, most reads coming from circRNA fragments will be attributed to their linear isoforms. Precise quantification of circRNAs is, thus, more complex. They are formed by a back-splicing process [34], causing the order of the exons on a given fragment to be different near the back-splicing site, also known as the back-splicing junction (BSJ), enabling the differentiation of circular from linear isoforms. Existing circRNAs detection tools like Circexplorer2 and CIRI2 use distinct strategies to detect BSJs [35,36], reporting counts that are not directly comparable with linear RNA counts, limiting comparisons between transcript types within datasets [37]. Similarly, the detection and quantification of fusion transcripts in short-read RNA-seq data requires specific tools [38,39] and are designed to identify and count fusion breakpoints, where exon sequences coming from two different genes are combined in a single fragment. Li et al. [40] and Wen et al. [41] implemented precise quantification of circRNAs concurrently with linear RNAs using a model-based framework. Rodosthenous et al. [31] showed profiling of linear long-coding and non-coding RNAs in plasma-derived EVs using an extended ncRNA annotation. We constructed a custom reference combining linear RNAs, an extended ncRNA annotation, circRNAs, and fusion transcripts relevant in the context of B-ALL. This approach enabled rapid, extensive, and comparable quantification of both linear and circular biotypes all within Salmon, a single existing model-based algorithm [42]. Counting circRNAs and fusion transcripts still relies on the presence of BSJs and fusion breakpoints and, thus, only reads containing such events can be attributed to those RNA biotypes. All other reads will be attributed to their regular linear counterparts, artificially increasing the ratio of linear to circular isoforms by an unknown amount. This inherent limitation of short-read RNA-seq will only be overcome by the use of full-length sequencing approaches.

Using our custom reference, we characterized the transcriptomic landscape of tumoral cells and IM-sEVs. Small EVs circulating in peripheral blood are secreted by most cells in the body [43], and while multiple studies have shown an overall increase in the concentration of sEVs in the blood of cancer patients vs. healthy donors [15,44,45], tumor-derived sEVs represent only a small fraction of all circulating vesicles [28]. The genes identified within patient-derived IM-sEVs may also implicate other concurrent mechanisms like immune function, acute injuries, or infection, whether directly related to the disease or not [46,47]. Moreover, in accordance with recent work by Padilla et al. [33], we found that non-coding RNAs in general, including both lncRNAs and circRNAs, were enriched in sEVs, suggesting potential mechanisms beyond stochastic diffusion involved in their export.

Comparing IM-sEVs RNA and tumoral RNA, only one gene detected in the IM-sEVs compartment was overexpressed in tumors of one subtype versus the other. Compared to controls, 23 genes nevertheless exhibited significant differential expression (9 up and 14 down) for tumors of both B-ALL subtypes. These findings support existing evidence that complex regulatory mechanisms govern the RNA transcript export into EVs [48]. Multiple studies have shown evidence for elevated EVs concentrations in the bloodstream of cancer patients [45,49,50], suggesting increased secretion by tumor cells or systemic factors affecting the whole organism (e.g., increased metabolism). RNA export to EVs has also been associated with key aspects of cancer biology, such as drug resistance [45,51], modulation of the tumor microenvironment [52,53], and the development of distant metastatic sites [54,55]. These individual mechanisms may be influenced by increased or decreased expression levels of specific transcripts, which may explain inverse correlations between enrichment in the vesicular compartment and levels in tumors.

Our approach allowed us to identify 102 potential circulating RNA biomarkers for two subtypes of childhood B-ALL in a subset of IM-sEVs enriched from patient peripheral blood. Most of these biomarker candidates (87 out of 102) were subtype-specific. The presence of a subset of those transcripts was also validated using RT-qPCR. The candidates comprised mRNAs, lncRNAs, circRNAs and pseudogenes, and differed from transcripts previously highlighted in tumoral RNA sequencing studies using BM biopsies as input material [6,8]. Of note, *ETV6::RUNX1* and *TCF3::PBX1* fusion transcripts, commonly used for subtype classification and monitoring in the current clinical practice [56,57], were not detected in patient blood IM-sEVs using either RNA-seq or RT-qPCR, which suggests that these fusion transcripts are not secreted into this compartment. However, we identified previously unknown B-ALL subtype-specific RNA transcripts in IM-sEVs, highlighting the important contribution of the extensive annotation constructed for this study. Working with only gencode annotation, none of the 10 circRNAs nor the 16 lncRNAs would have been detected or identified. Our results suggest that IM-sEVs shuttled in peripheral blood contain a subtype-specific signature linked to childhood B-ALL, distinct from the signature known from studies examining transcriptomes derived from BM biopsies.

To gain further insight into the functional implications of the RNA biomarker candidates identified here, we performed correlation-based WGCNA and FGSEA computational analyses on tumoral datasets. This analysis supported the relationship between RNA biomarker subset candidates and the presence of B-ALL or B-ALL subtypes, highlighting their potential involvement in multiple cancer-related pathways. When looking at the presence of B-ALL, both subtypes included the 10 top enriched GO terms; almost all point at mechanisms of DNA replication and cell-division. However, when looking at one subtype versus the other, there is no evident link with general cancer mechanisms. Module B6, associated with the *ETV6::RUNX1^+^* subtype, is enriched for genes linked to bone morphology. A subset of B-ALL patients present with bone lesions at diagnosis [58] and multiple studies have linked bone involvement at diagnosis with a better prognosis [59,60]. Module B3, associated with the *TCF3::PBX1^+^* subtype, is enriched for genes linked to the regulation of excitatory postsynaptic potential and the postsynaptic membrane. Central nervous system involvement is also a known feature of childhood B-ALL with a strong prognostic value [2,61]. The subtype-specific presence of these transcripts in circulating EVs from childhood B-ALL patients suggests that they could be involved in similar mechanisms and should be further studied in the context of childhood B-ALL. Among the 67 mRNAs out of the 102 potential biomarkers, 29 are encoding for ribosomal proteins of the RPL and RPS families. Jenjaroenpun et al. [62] suggested that certain sEVs-shuttled mRNAs may influence ribosomal activity in recipient cells, possibly augmenting the translation of co-encapsulated transcripts. Moreover, previous research has associated ribosomal protein expression with cellular proliferation in cancer [63,64,65]. Caron et al. reported an inverse correlation between the expression of ribosomal protein genes and the developmental state of leukemia cells, including *ETV6::RUNX1*^+^ samples, with potential subtype specificity [66]. While we did not observe significant differential expression of ribosomal proteins in *ETV6::RUNX1*^+^ and *TCF3::PBX1^+^* BM biopsies, our findings underscore the need for further investigation into subtype-specific export of ribosomal protein genes within circulating sEVs derived from the blood of childhood B-ALL patients. Using FGSEA, we observed two main functionally distinct subclusters within each childhood B-ALL molecular subtype. For the *ETV6::RUNX1^+^* subtype clusters A and B (Figure 5E and Appendix A), they are mostly defined by correlation with the oxidative phosphorylation, MYC targets, and E2F targets gene sets. For the *TCF3::PBX1^+^* subtype clusters C and D (Figure 5E and Appendix A), they are mostly defined by correlation with the Heme metabolism, MTORC1 signalling, MYC targets, protein secretion, and G2M checkpoint gene sets. These observations show that RNA transcripts packaged in blood-derived IM-sEVs from childhood B-ALL patients are associated with a variety of cellular pathways. These associations differ between the two distinct molecular subtypes studied (*ETV6::RUNX1^+^* and *TCF3::PBX1^+^*), pointing at potential subtype-specific signatures present in sEV RNA.

## 4. Materials and Methods

**Patient samples.** Our study cohort consisted of 24 B-ALL from the Quebec childhood ALL cohort (QcALL) diagnosed in the Division of Hematology-Oncology at the Sainte-Justine Hospital (Montreal, QC, Canada) [67]. A subset of 10 patients out of the 24 also had their blood EVs purified using size selection and immune purification (IM-sEVs) and IM-sEVs RNA sequenced for this study (Table 1). BM biopsies and peripheral blood drawn in K2 EDTA tubes were collected at diagnosis. Within 1 h of collection, whole-blood samples were layered over a Ficoll-Paque density gradient medium (MilliporeSigma, St. Louis, MO, USA) and centrifuged at 400× *g* for 30 min at room temperature (RT) without applying brakes. The top plasma layer was collected and centrifuged a second time at 2000× *g* for 10 min at RT. BM biopsies were also layered over a Ficoll-Paque density gradient medium (Sigma) and centrifuged at 400× *g* for 30 min at room temperature (RT) without applying brakes. The cellular fraction was collected, washed twice with PBS, and pelleted. Both BM-biopsy-derived cell pellets and plasma samples were kept frozen at −80 °C until further processing. We used 3 samples consisting of CD10^+^/CD19^+^ B-cells as controls. They were purified from fresh human cord blood by positive selection using CD10^+^ and CD19^+^ magnetic microbeads (Miltenyi Biotec, Gaithersburg, MD, USA) [8]. The Sainte-Justine institutional review board approved the research protocol, and written informed consent was obtained from all participants and their parents or legal guardians. This study was performed in accordance with the Declaration of Helsinki.

**Cell culture.** REH (*ETV6::RUNX1*^+^ Human B cell precursor leukemia; ATCC CRL-1567) and 697 (*TCF3::PBX1*^+^ Human B cell precursor leukemia; DSMZ ACC-42) cells were first cultured in RPMI 1640 medium (Wisent, Saint-Jean-Baptiste, QC, Canada) supplemented with 10% heat-inactivated fetal bovine serum (FBS) (Wisent) and 1% penicillin/streptomycin (Wisent). They were then transferred gradually to 100% X-VIVO 15 Serum-free Hematopoietic Cell Medium (Lonza, Basel, Switzerland) over the course of 4 passages. Both cell lines were cultured in a 37 °C incubator with 5% CO_2_. Conditioned cell culture medium (CCM) was collected when cell concentrations reached 10^6^ cells/mL and apoptosis was kept under 5%.

**EVs fractionation and purification.** For patients’ samples, 900 µL of thawed plasma was centrifuged at 10,000× *g* for 10 min at room temperature and filtered using a 1 mL syringe and a 0.8 µm cellulose acetate filter membrane (GVS, Bologna, Italy) following the recommendations of Gaspar et al. [24]. For the cellular models, 45 mL of CCM was collected at 70% cell confluency and centrifuged at 300× *g* for 10 min at room temperature to remove suspended cells, followed by a second round of centrifugation at 2500× *g* for 10 min at room temperature to remove large cell debris. The supernatant was then collected and concentrated down to 900 µL by ultrafiltration using Amicon Ultra-15 Centrifugal Filter units (MilliporeSigma) with 100 kDa molecular weight cut-off. Either 900 uL of filtered plasma or 900 µL of concentrated CCM was then used as input for EVs isolation by SEC using qEV original 70 nm columns (Izon Sience, Christchurch, New Zealand). For each sample, 4 × 500 µL fractions were collected, centrifuged at 20,000× *g* to remove larger microvesicles (MVs). The supernatant containing sEVs was then concentrated down to 0.5 mL using Amicon Ultra-0.5 Centrifugal Filter units (MilliporeSigma) with 30 kDa molecular weight cut-off. Each concentrated sample was directly processed with the EasySep Human Pan-Extracellular Vesicle Positive Selection Kit (Stemcell, Vancouver, BC, Canada) to purify sEVs bearing various combinations of the CD9, CD63, and CD81 surface proteins, following the manufacturer’s protocol. We refer to these CD9^+^/CD63^+^/CD81^+^ sEVs population as IM-sEVs. After the final washing step, the bead bound IM-sEVs were lysed in buffer RL (Norgen, Thorold, ON, Canada) or RIPA buffer (MilliporeSigma) and stored at −80 °C before further processing. Figure 1A,B show schematics summarizing both protocols.

**Tunable resistive pulse sensing (TRPS).** TRPS measurements were conducted using a qNano instrument (Izon Science, Christchurch, New Zealand). Various polyurethane nanopores, appropriately stretched (NP100, NP150, NP200, Izon Science), and the matching diluted calibration polystyrene beads (CPC100, CPC200, Izon Science) were used for sEVs measurements. All samples were diluted in PBS with 25 mM Trehalose and 0.05% Tween20. Each measurement was performed at two different pressures (5 and 10 mbar). Data were processed and analyzed using the Izon Control Suite software v3.3.2.2001.

**IM-sEVs purity measurements.** Purified IM-sEVs lysed in RIPA buffer were assessed for the concentration of four EV-specific protein markers (CD9, CD63, CD81, and syntenin-1), one negative control (cytochrome-c), and five lipoproteins (Apo-A1, Apo-A2, Apo-B, Apo-C2, and Apo-E) using the Exosome Characterization 6-Plex Human ProcartaPlex Panel and the ProcartaPlex Human Apolipoprotein 5-Plex Panel (Invitrogen, Waltham, MA, USA) on a CS 1000 Autoplex Analyzer (PerkinElmer, Waltham, MA, USA). Both kits include positive samples of known concentration for each measured protein standard, and background is established on the assay diluent, enabling precise and absolute quantification of protein concentrations in unknown samples.

**Cells and EVs RNA isolation.** Total RNA from whole-cell pellets and bead-bound IM-sEVs (CCM- and blood-derived) was extracted using the Total RNA Purification Micro Kit (Norgen), following the manufacturer’s instructions, with added on-column DNAse digestion (Norgen). For the patient’s tumoral samples (BM biopsies) and B-cell controls, total RNA was previously extracted from leukemic cells using the mini AllPrep DNA/RNA kit (Qiagen, Hilden, Germany). RNA yield and quality was assessed by electrophoresis on a 2100 bioanalyzer with the RNA 6000 nano or pico kits (Agilent, Santa Clara, CA, USA).

**Library preparation and sequencing.** Library preparation was conducted using the SMARTer Stranded Total RNA-Seq Kit v2—Pico Input Mammalian kit (Takara, San Jose, CA, USA). For patient’s IM-sEVs samples, 1 ng total RNA was used as input. For the cellular models, 10 ng total RNA was used as input for both IM-sEVs and whole-cell extracts. All samples were subjected to RNA fragmentation for 2 min at 94 °C followed by cDNA synthesis, a 5-cycle indexing PCR, ribosomal cDNA depletion, and a 14-cycle enrichment PCR. Libraries were quantified on a 2100 bioanalyzer with the DNA High-Sensitivity kit (Agilent) and sequenced (paired-end 2 × 75 bp or 2 × 150 bp) on a Nextseq 500 (Illumina, San Diego, CA, USA) to 100 million read pairs per library. For patient tumor samples and B-cell controls, TruSeq Stranded Total RNA libraries were previously prepared using the Ribo-Zero Gold Kit (Illumina), according to the manufacturer’s protocol, and sequenced (paired-end 2 × 75 bp or 2 × 100 bp) on HiSEQ 2500 or HiSeq 4000 systems (Illumina).

**Bioinformatic analyses.** To quantify a wide variety of coding and non-coding linear RNAs, specific fusion transcripts, and circRNAs using a single program, a custom reference was created by merging Gencode V29, LNCipedia 5.2, circRNAs detected with CIRCexplorer2 [35] and CIRI2 [36], and fusion transcripts detected with STAR-Fusion [39] and Arriba [38]. Reads obtained from the SMARTer Stranded libraries were quality trimmed using TrimGalore 0.6.6. The Ribo-Zero Gold libraries of tumoral samples were analyzed without any pre-processing. All libraries were quasi-mapped to the transcriptome using Salmon 1.9.0 [42] with the custom reference, using the following parameters: -l ISR—minScoreFraction 0.55 –seqBias –gcBias –dumpEq. Raw read counts for all transcripts derived from Salmon were converted to gene-level expression with the tximport 1.24.0 R package [68], followed by normalization with DESeq2 1.36.0 [69], and aggregation by gene type to assess the overall RNA composition of the samples. DESeq2 was used to identify genes significantly enriched in IM-sEVs and highlight the proportions of each gene type in the enriched gene population. Venn diagrams were drawn using the VennDiagram 1.7.3 R package. Correlation and ontology analyses were conducted using the WGCNA, fgsea, and pathfindR R packages [70,71,72]. Complete procedures are available in the Appendix A.

**RNA-seq results validation.** Due to very limited availability of childhood B-ALL patients’ plasma at diagnosis (900 µL only per patient), we used most of the RNA extracted from IM-sEVs for sequencing. Remaining patient RNA thus had to be pooled by B-ALL subtype (5 patients each) to obtain enough RNA for reverse transcription quantitative polymerase chain reaction (RT-qPCR) validation. While this approach may not be optimal for highlighting individual patient differences, it offers the advantage of reducing noise at the subtype level and strengthening observations linked to subtype-specific risk stratification. Starting with 2 ng RNA from each pool, we generated double-stranded cDNA by reverse transcription (RT) via random priming followed by template switching with the Pico v2 SMART adapter from the Pico Input Mammalian RNA-seq kit (Takara), omitting the RNA fragmentation step. We then conducted 11 cycles of PCR pre-amplification targeting sequences introduced on every cDNA molecule at the RT step. Gene expression was quantified by RT-qPCR in triplicate. Divergent primers for circRNAs were designed using a combination of CircInteractome [73] and the NCBI Primer designing tool. Primers for linear RNAs were designed using the NCBI Primer designing tool. All gene-specific primer sequences are listed in Appendix A.

**Statistical analyses.** For all reported gene expression values from tumoral and IM-sEVs RNA-seq, statistical differences were assessed using the Mann–Whitney U two-sided test using the rstatix 0.7.2 R package. P-values were adjusted with the Bonferroni correction method. Adj. *p* < 0.05 was considered statistically significant. For weighted gene correlation network analysis (WGCNA), links between modules and traits were assessed by Pearson correlation. For gene set enrichment analysis, correlations were assessed by the FGSEA package using an adaptive multi-level split Monte Carlo scheme described by Korotkevich et al. [71].

We provide promising insights into the use of long circulating RNAs packaged in EVs as non-invasive biomarkers for childhood B-ALL.

## 5. Conclusions

Our study provides a first description of the long transcriptome landscape in blood-derived IM-sEVs of two childhood B-ALL subtypes with distinct risk profiles. We highlighted the potential of circulating, EV-shuttled, mRNAs, lncRNAs, circRNAs, and pseudogenes as biomarkers for risk-based patient stratification and explored complex mechanisms underlying the export of RNA transcripts in EVs and functional implications in cancer-related pathways. EV RNAs offer a promising non-invasive avenue for risk stratification at diagnosis and should also be studied for continuous patient monitoring throughout treatment. As in the case of MRD follow-ups from peripheral blood [13], utilizing EV RNAs may alleviate the traumatic burden on patients by removing the need for frequent BM biopsies and enabling the collection of samples at higher frequencies. Moreover, analyses based on peripheral blood samples can be conducted days before tumoral material becomes available, facilitating rapid risk assessment and potentially expediting the initiation of personalized treatment strategies. Therefore, the integration of EV RNA-based liquid biopsies into clinical practice holds the potential to significantly reduce the time required for precise risk stratification from weeks to days, which may improve patient outcomes [74]. Our study is the first to underscore the potential of long sEV-derived RNAs as biomarkers of B-ALL subtypes.

However, several limitations should be acknowledged. Given that childhood B-ALL is a rare pediatric disease, our sample size was limited to 24 patients, covering two subtypes of B-ALL, with plasma samples available at diagnosis for only 12 patients. This small sample size may affect the generalizability of our findings, and larger cohorts will be necessary to validate the identified RNA biomarker candidates and establish their clinical utility.

While the study identifies a potential subtype-specific signature, we did not explore correlations between circulating RNAs and other important clinical outcomes, such as therapeutic response and risk of relapse, due to the lack of available datasets. Future investigations should include patient samples taken at various timepoints during treatment and long-term follow-up to enhance the prognostic value of circulating RNAs.

The methodology employed for EV isolation and RNA sequencing is innovative, allowing us to work with very limited quantities of patient samples. However, it presents challenges for widespread application in clinical settings, as these techniques require specialized equipment and expertise, potentially limiting their accessibility for routine clinical use. We anticipate improvements in the profiling of full-length EV-derived RNA transcripts once long read sequencing methods become viable for low RNA input.

While the integration of EV-derived RNA biomarker analysis into clinical practice is promising, research protocols can be optimized to further refine biomarker panels. These panels could then be adapted to more accessible techniques in the clinic, such as qPCR-based detection or targeted sequencing. Additionally, there is a need for streamlined methods for EV purification in the clinical setting to facilitate this transition.

## Figures and Tables

**Figure 1 ijms-26-03956-f001:**
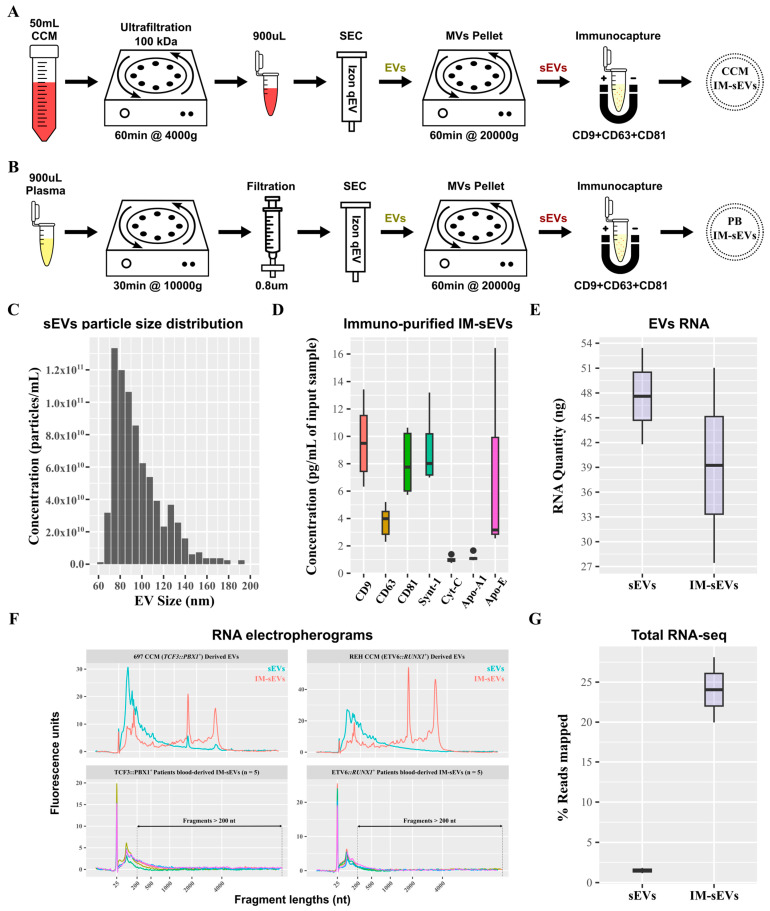
Purification of sEV fractions from peripheral blood and CCM. Schematics of the optimized protocols for purification of EVs from (**A**) CCM and (**B**) plasma from peripheral blood (PB). (**C**) Histogram showing the particle size distribution of the sEV fraction measured by TRPS. Boxplots showing (**D**) protein concentration of exosomal proteins (CD9, CD63, CD81, Synt-1), lipoproteins (Apo-A1, Apo-E), and negative control (Cyt-C) after immunoprecipitation of sEVs using magnetic beads functionalized with CD9/CD63/CD81 antibodies. (**E**) RNA concentration in sEV fractions purified from CCM measured by electrophoresis. (**F**) Electropherograms of RNA samples from EV fractions (sEVs in blue and IM-sEVs in red) purified from CCM of 697 (*TCF3::PBX1^+^*) and REH (*ETV6::RUNX1^+^*) cell lines and patient-blood-derived IM-sEVs (n = 5 for *TCF3::PBX1^+^* and n = 5 for *ETV6::RUNX1^+^*, each color representing a single patient sample). Fragments > 200 nt are highlighted for patient-blood-derived IM-sEVs. (**G**) Boxplot showing the percentage of RNA-seq read mapping to the transcriptome for EV fractions purified from CCM.

**Figure 2 ijms-26-03956-f002:**
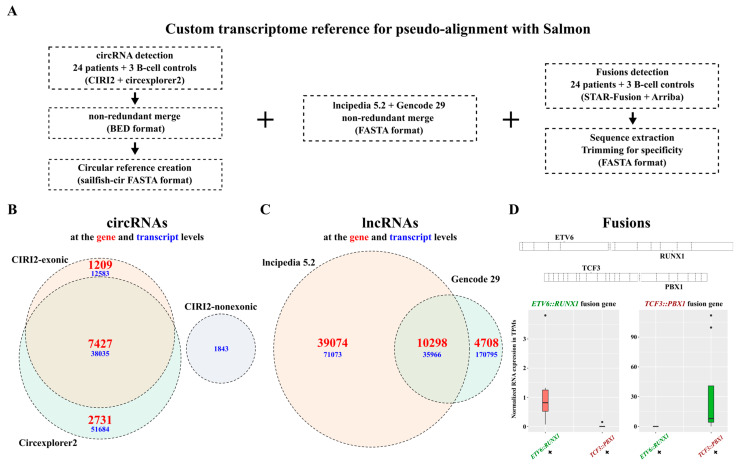
Custom transcriptome reference for extensive RNA biotypes quantification in B-ALL RNA-seq datasets. (**A**) Custom transcriptome assembly generated by merging circRNAs detected with CIRI2 and circexplorer2, lncRNAs from lncipedia 5.2, and Gencode V29, and *ETV6::RUNX1* and *TCF3::PBX1* fusions sequences detected with STAR-Fusion and Arriba. (**B**) Venn diagram showing the number of circRNAs included in the reference, detected from BM biopsies (n = 24), CD10^+^/CD19^+^ B-cells derived from healthy cord blood (n = 3), and blood-derived exosomes (n = 10) from our patient cohort, as well as 697 and REH cells and CCM-derived exosomes. (**C**) Venn diagram showing the number of lncRNAs included in the reference with a non-redundant merge of gencode 29 and lncipedia 5.2 (circle sizes are proportional to the gene values). (**D**) Quantification of *ETV6::RUNX1* and *TCF3::PBX1* fusion genes in RNA-seq data from patient BM biopsies (n = 24) and 697 and REH whole-cell extracts using pseudoalignment to demonstrate specificity.

**Figure 3 ijms-26-03956-f003:**
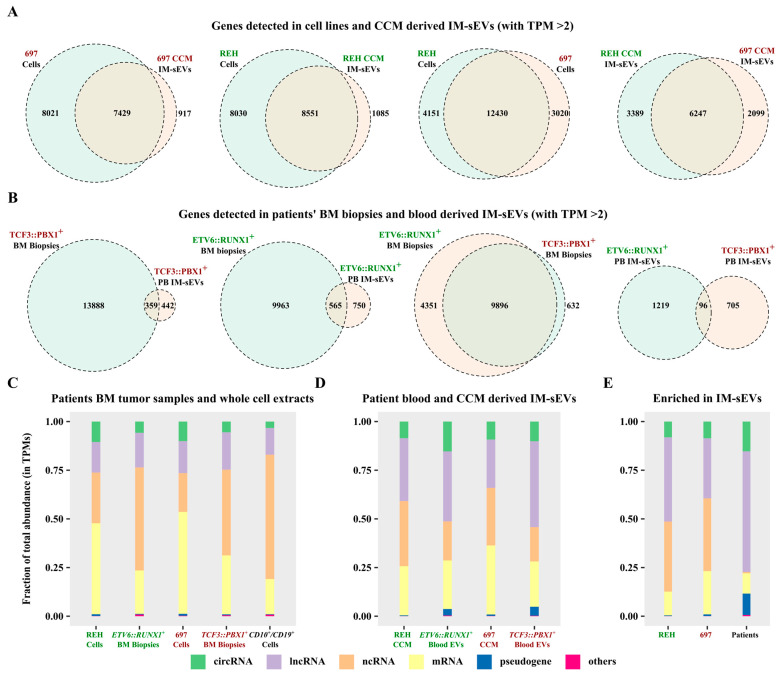
Characterization of the long transcriptome in circulating IM-sEVs of childhood B-ALL patients and cell lines. (**A**) Two-way Venn diagrams showing genes detected with a TPM > 2 threshold in RNA-seq data from whole-cell extract and IM-sEVs purified from CCM of the REH (*ETV6::RUNX1^+^*) and 697 (*TCF3::PBX1^+^*) cell lines. (**B**) Two-way Venn diagrams showing genes detected with a TPM > 2 threshold in RNA-seq data from BM tumors (n = 24) and IM-sEVs purified from the peripheral blood of *ETV6::RUNX1^+^* and *TCF3::PBX1^+^* patients (n = 10). Stacked bar plots showing the RNA biotypes distributions as percentages of total TPMs for (**C**) whole-cell extract of REH and 697 cell-lines, BM tumor samples from *ETV6::RUNX1^+^* (n = 12) and *TCF3::PBX1^+^* (n = 12) patients, and CD10+/CD19+ B-cells (n = 3) purified from healthy human cord blood as controls. (**D**) CCM-purified IM-sEVs of REH and 697 cell lines and peripheral blood IM-sEVs from *ETV6::RUNX1^+^* (n = 5) and *TCF3::PBX1^+^* (n = 5) patients. (**E**) Transcripts enriched in the IM-sEVs compartment for *ETV6::RUNX1^+^* and *TCF3::PBX1^+^* patients combined and for REH and 697 CCM as compared with BM tumors and whole-cell extracts.

**Figure 4 ijms-26-03956-f004:**
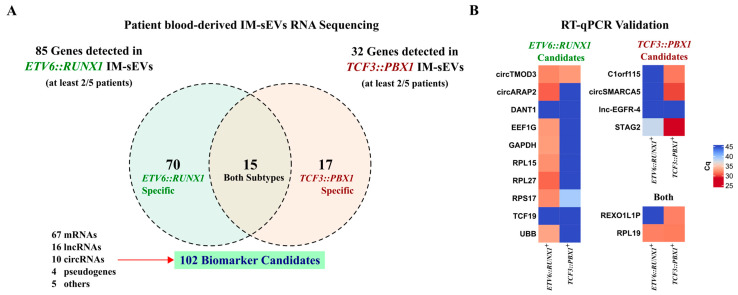
Identification of circulating RNA biomarker candidates in childhood B-ALL blood-derived IM-sEVs. (**A**) Venn diagram showing genes detected in at least 2/5 IM-sEVs samples for *ETV6::RUNX1^+^* (n = 5) and *TCF3::PBX1^+^* (n = 5) patients. 102 EV RNA biomarker candidates: 70 specific to *ETV6::RUNX1*, 17 specific to *TCF3::PBX1* and 15 in both subtypes (**B**) Heatmap showing the expression levels reported as Cq values for selected RNA biomarker candidates (16/102) quantified by RT-qPCR in blood-derived IM-sEVs from *ETV6::RUNX1^+^* (n = 5) and *TCF3::PBX1^+^* (n = 5) patients, starting from 2 ng RNA pooled by subtype.

**Figure 5 ijms-26-03956-f005:**
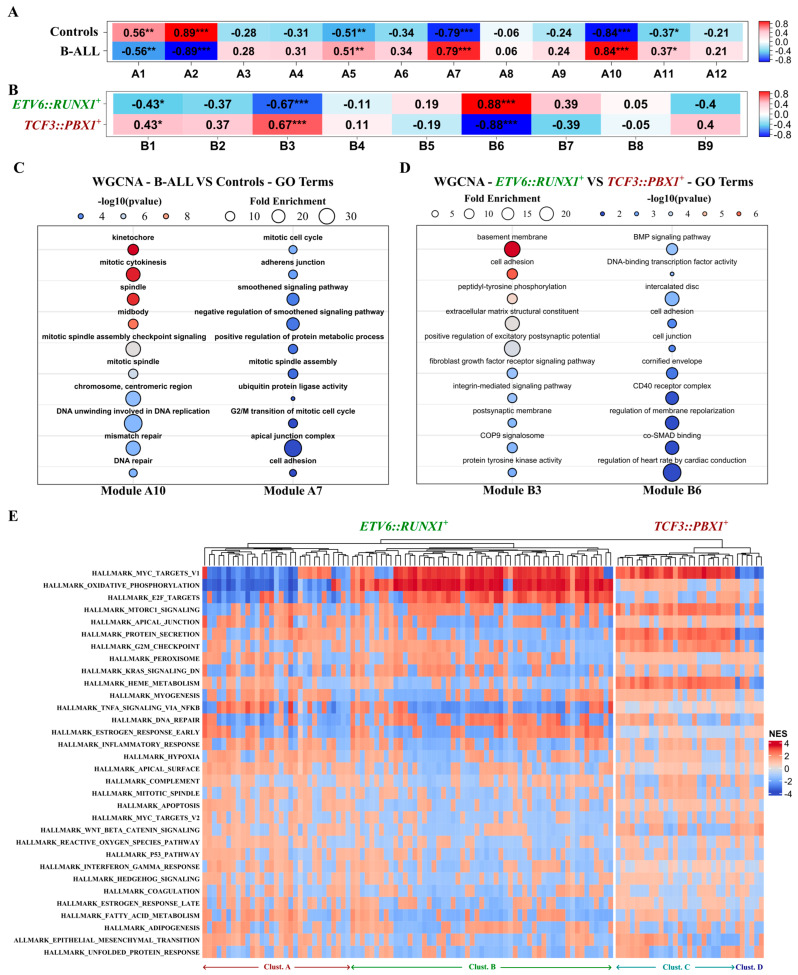
Putative functions of the identified RNA biomarker candidates in childhood B-ALL tumorigenesis. WGCNA analysis conducted on the top 5000 variable genes, including the 102 potential biomarker candidates on two distinct datasets. (**A**,**B**) Heatmap depicting the correlation between WGCNA identified modules and disease-associated traits Pearson correlation; * *p*-value < 0.05; ** *p*-value < 0.01; *** *p*-value < 0.001. (**A**) Combined (n = 24) B-ALL patients’ tumors from BM biopsies (*ETV6::RUNX1^+^* and *TCF3::PBX1^+^*) versus CD10+/CD19+ B-cells (n = 3) purified from healthy human cord blood as controls; (**B**) *ETV6::RUNX1^+^* (n = 12) versus *TCF3::PBX1^+^* (n = 12) tumors. (**C**,**D**) Bubble plots showing the top 10 GO terms enriched in (**C**) the A10 and A7 modules correlated with B-ALL (*ETV6::RUNX1^+^* and *TCF3::PBX1^+^* tumors combined, n = 24) and (**D**) the B3 and B6 modules, respectively, correlated with the *ETV6::RUNX1^+^* (n = 12) and *TCF3::PBX1^+^* (n = 12) disease subtypes. (**E**) Heatmap depicting the correlation between the expression of the 102 identified childhood B-ALL RNA biomarker candidates and 33/50 MSigDB hallmark gene sets. Correlations assessed by the FGSEA package using an adaptive multi-level split Monte Carlo scheme.

**Table 1 ijms-26-03956-t001:** Patient cohort statistics.

	Blood-Derived IM-sEVs RNA-seq	BM-Biopsy-Derived Tumoral RNA-seq
No. of individuals	10 out of 24	24
Mean age at diagnosis ± SD, y	8.9 ± 4.5	7.3 ± 4.6
Sex		
Males, n (%)	4 (40)	8 (33.3)
Females, n (%)	6 (60)	16 (66.6)
Genetic alterations		
*ETV6::RUNX1*, n (%)	5 (50)	12 (50)
*TCF3::PBX1*, n (%)	5 (50)	12 (50)

SD: Standard deviation.

## Data Availability

All raw and processed sequencing data generated in this study have been submitted to the NCBI Gene Expression Omnibus (GEO; https://www.ncbi.nlm.nih.gov/geo/) under accession number GSE245139.

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
