# Peer review of "Long Circulating RNAs Packaged in Extracellular Vesicles: Prospects for Improved Risk Assessment in Childhood B-Cell Acute Lymphoblastic Leukemia"

_ijms, 2025, doi:10.3390/ijms26093956_

Round 1
Reviewer 1 Report
Comments and Suggestions for Authors
GENERAL COMMENTS
This is a difficult to read manuscript, analyzing the detection and potential prognostic importance of circulating, long non-coding RNAs in pediatric patients of B-ALL of two specific cytogenetic types. The idea is great but the methodology rather too complicated to be applied in everyday clinical practice. The authors have collected, both, bone marrow and peripheral blood from 24 patients, as mentioned in materials and methods section. Following isolation procedures authors obtained immuno-purified extracellular vehicles (IM-EVs) from peripheral blood. At this point, the authors should present clearly their concept of interpreting NGS RNA data from bone marrow cells, against IM-EVs derived from peripheral blood plasma, including an appropriate literature support. More precisely, can the peripheral blood plasma exosomes (IM-EVs) mirror molecular changes, been occurring in the bone marrow? In other words, are the circulating exosomal RNAs relevant as early indicators of disease state and subtype? Regarding the methodology, although appropriately designed, I think that the authors should comment on why they did not choose to perform Third Generation Sequencing and to harvest on a more practical and potentially applicable way the whole RNA-ome of the targeted biological material. Otherwise, the manuscript deserves to be published, but there is need to be improved, not only on the specific issues mentioned below, but also in several other points at which the description of the results sounds clearly complicated, particularly by clinicians, with the use of abbreviations and acronyms, which are not commonly used.
SPECIFIC ISSUES FOR CLARIFICATION
Abstract, line 20: The authors need to make clear what they imply with the phrase “despite their potential for patient classification.” Which is this potential and how it could affect patient classification?
Abstract, line 23: “Long transcriptome sequencing”: This expression should be rephrased since the authors have performed RNA-seq (NGS) combined with downstream bioinformatics analysis to improve the efficient detection of long RNA molecular species, instead of Long-Read RNA Sequencing (Third-Generation sequencing).
Introduction, line 69: Please, define what is “long RNA”
Introduction, line 78: The authors should define the term “long transcriptome”. Do they mean transcripts longer than miRNAs which have been so far the main target included in EVs’ cargo? Do they mean coding and non-coding transcripts longer than 200nt? Electropherograms (Figure 1F) depict different sizes of isolated RNA molecules derived from cell lines (EVs) and RNA derived from patients (IM-sEVs), however fragment lengths are not referenced in the text or in the figure legend.
The manuscript title also includes the term “long”, which needs careful clarification within the text, otherwise it should be omitted.
The authors should explain, particularly for the circRNA validations the following:
- How have they excluded any false positives? - Is there any possibility of NGS detected circRNAs to be artifacts due to rRNA contamination?
-Experimental validation for detected circRNAs has been performed with RT-PCR utilizing divergent primers however, RNase, R treatment to remove linear RNAs is not mentioned.
Discussion lines 327-330: “We first observed that the long RNA cargo in CCM derived IM- sEVs is highly representative of the expressed transcriptome in the secreting cells. Conversely, blood derived IM-sEVs contained significantly less genes than the tumoral transcriptome”: This comparison is irrelevant since authors haven’t performed correlations between tumor cells against IM-EVs, both deriving from the bone marrow tissue, where the exosomal RNA signatures would reflect the microenvironment from the bone marrow. This kind of correlation could lead to comparable results as the RNA cargo in CCM derived IM- sEVs and endogenous RNA species from the respective cells (cell lines).
Materials and Methods, line 413: “BM biopsies and plasma samples were kept frozen until further processing”: The authors should describe here the freezing conditions prior to RNA extraction, to avoid sample degradation.
Reviewer 2 Report
Comments and Suggestions for Authors
The authors developed a novel method – immune-purification (IM) followed with traditional size exclusion chromatography (SEC) – to isolate long circulating RNAs in extracellular vesicles (EVs). Then the authors applied this novel method to isolate IM-sEV-derived long circulating RNAs from patients samples or cell line models with either ETV6::RUNX1 or TCF3::PBX1 translocation. The quality of RNA derived from IM-sEV is comparable from general sEV but showed slightly higher sequencing read mapping. Then they applied RNA sequencing on both IM-sEV-derived RNA and cell-derived RNAs from both primary patient samples and cell culture samples in both genotypes. The long circulating RNA landscape in IM-sEV is quite different from RNA expression within cells. The authors called 102 long circulation RNAs that expressed in IM-sEVs from either ETV6::RUNX1 or TCF3::PBX1 or both genotypes as a list of biomarkers for downstream analysis. They applied these 102 biomarkers in two sets of data derived from primary patients and healthy controls to perform a weighed gene co-expression network analysis (WGCNA). The findings are that 1) these 102 EV-long circulating RNA biomarkers are associated with more advanced stage of ETV6::RUNX1 B cell acute lymphoid leukemia (B-ALL) and 2) some of these biomarkers is better correlates with ETV::RUNX1 chromosomal alteration. Overall, this study carry significant impact for revealing the prognostic biomarkers for stratifying B-ALL in a non-invasive way. However, this manuscript failed in addressing some major questions.
- The analysis is circular by itself. Those 102 biomarkers are derived from cancer cells (either primary patient sample or cancer cell lines, the authors need to note that better in the manuscript text or figure legend), so it is not surprising to see it correlate closer to more advanced tumor cells compared to healthy control. And since most of the biomarkers are derived from ETV::RUNX1 samples, so it correlates better with ETV::RUNX1 samples compared to TCF3::PBX1 samples.
- The authors should apply these 102 biomarkers from a larger publicly available data base that not only include these two chromosome alterations and see if these biomarkers are associated with survival, response to therapy.
- The title is prospects for improved risk assessment, but 1) there is no specific long circulating RNA signatures identified to be applied in clinical for risk prediction; 2) the authors did not preform specific correlation analysis between these biomarkers with certain risk (relapse, not responding to certain therapy and etc.)
- The authors did not specify is the 102 RNA candidates are from patient data or cell line data or both.
- The overall English writing need to be improved. For example, the sentence in line 91 “Plasma derived IM-sEVs were almost completely depleted from lipoproteins compared to plasma input samples (Figure S1)”. It is hard to get the information from figure S1, isn’t the authors trying to harvest IM-sEVs from plasma and blood cells? And the next sentence after this sentence is also confusing, please rewrite for clarification. And the paragraph from line 212 to 217 is hard to process “ … no significant differences in expression levels in tumor samples for the 102 putative RN candidates. … However, … 9 candidates exhibited significantly higher expression in tumors…”. Please address the clarification issue throughout the manuscript.
